# Current Applications and Challenges of Next-Generation Sequencing in Plasma Circulating Tumour DNA of Ovarian Cancer

**DOI:** 10.3390/biology13020088

**Published:** 2024-01-31

**Authors:** Ricardo Roque, Ilda Patrícia Ribeiro, Margarida Figueiredo-Dias, Charlie Gourley, Isabel Marques Carreira

**Affiliations:** 1Cytogenetics and Genomics Laboratory, Institute of Cellular and Molecular Biology, Faculty of Medicine, University of Coimbra, 3004-504 Coimbra, Portugal; 2Centre of Investigation on Environment Genetics and Oncobiology (CIMAGO), Institute for Clinical and Biomedical Research (iCBR), Faculty of Medicine, University of Coimbra, 3004-504 Coimbra, Portugal; 3Portuguese Institute of Oncology of Coimbra, 3000-075 Coimbra, Portugal; 4Faculty of Medicine, Gynecology Department, University of Coimbra, 3004-504 Coimbra, Portugal; 5Coimbra Academic and Clinical Centre, 3000-370 Coimbra, Portugal; 6Gynecology Department, Hospital University Centre of Coimbra, 3004-561 Coimbra, Portugal; 7Nicola Murray Centre for Ovarian Cancer Research, Cancer Research UK Scotland Centre, Institute of Genetics and Cancer, University of Edinburgh, Edinburgh EH4 2XU, UK

**Keywords:** ovarian cancer, liquid biopsy, circulating tumour DNA, next-generation sequencing

## Abstract

**Simple Summary:**

Circulating tumour DNA (ctDNA) corresponds to fragments of cancer genetic material circulating in body fluids, such as blood. Its isolation and analysis have made it a prominent biomarker in all types of human cancer. Ovarian cancer needs better biomarkers to help predict prognosis, treatment response, and early recurrence, and therefore there is an increasing interest in determining the real utility of ctDNA in each phase of the disease. With the advent of next-generation sequencing, ctDNA-based studies have greater potential to facilitate monitoring tumour genetic alterations across all metastatic sites and longitudinally through time, overcoming the need for an invasive tissue biopsy. This follow-up can be informative for clinicians regarding resistance mechanisms and new therapeutic options for cancer treatment. Therefore, in this review, we will address and group the current knowledge on ctDNA as a biomarker in ovarian cancer, focusing on studies using sequencing methods. We aimed to identify the paucity of scientific knowledge on the subject and highlight more adequate methodologies for biomarker research in order to motivate and guide future studies in the field.

**Abstract:**

Circulating tumour DNA (ctDNA) facilitates longitudinal study of the tumour genome, which, unlike tumour tissue biopsies, globally reflects intratumor and intermetastatis heterogeneity. Despite its costs, next-generation sequencing (NGS) has revolutionised the study of ctDNA, ensuring a more comprehensive and multimodal approach, increasing data collection, and introducing new variables that can be correlated with clinical outcomes. Current NGS strategies can comprise a tumour-informed set of genes or the entire genome and detect a tumour fraction as low as 10^−5^. Despite some conflicting studies, there is evidence that ctDNA levels can predict the worse outcomes of ovarian cancer (OC) in both early and advanced disease. Changes in those levels can also be informative regarding treatment efficacy and tumour recurrence, capable of outperforming CA-125, currently the only universally utilised plasma biomarker in high-grade serous OC (HGSOC). Qualitative evaluation of sequencing shows that increasing copy number alterations and gene variants during treatment may correlate with a worse prognosis in HGSOC. However, following tumour clonality and emerging variants during treatment poses a more unique opportunity to define treatment response, select patients based on their emerging resistance mechanisms, like BRCA secondary mutations, and discover potential targetable variants. Sequencing of tumour biopsies and ctDNA is not always concordant, likely as a result of clonal heterogeneity, which is better captured in the plasma samples than it is in a large number of biopsies. These incoherences may reflect tumour clonality and reveal the acquired alterations that cause treatment resistance. Cell-free DNA methylation profiles can be used to distinguish OC from healthy individuals, and NGS methylation panels have been shown to have excellent diagnostic capabilities. Also, methylation signatures showed promise in explaining treatment responses, including BRCA dysfunction. ctDNA is evolving as a promising new biomarker to track tumour evolution and clonality through the treatment of early and advanced ovarian cancer, with potential applicability in prognostic prediction and treatment selection. While its role in HGSOC paves the way to clinical applicability, its potential interest in other histological subtypes of OC remains unknown.

## 1. Introduction

Ovarian cancer (OC) is the 7th most frequent gynaecological malignancy worldwide and the main cause of gynaecological cancer death [1,2]. OC is of epithelial origin in 90% of cases, and these can be classified into five different histological subgroups based on the World Health Organization’s (WHO) current classification: high-grade serous ovarian carcinoma (HGSOC), endometrioid carcinoma, clear-cell carcinoma, low-grade serous carcinoma, and mucinous carcinoma [3]. Most cases are diagnosed at advanced stages with peritoneal involvement, indicating poor overall survival (OS), despite the best therapeutic efforts [1,4]. However, different subtypes have diverse molecular and phenotypical behaviours, as well as distinct prognosis and treatment options [1].

HEA-4 and CA-125 are the two clinically useful serum protein biomarkers for OC. Only CA-125 is approved for evaluating treatment response and disease recurrence [5,6]. The absence of higher-sensitivity biomarkers capable of early detection and prognostication remains an area of need in the management of EOC [1,4,5,7,8]. In numerous cancers, cell-free DNA (cfDNA) has shown promise in predicting prognosis, assessing treatment response, and recurrence detection [4,7,9]. Table 1 compares CA-125 and ctDNA as biomarkers of OC.

Cell-free DNA (cfDNA) is released by both malignant and healthy cells through apoptosis and other cell death mechanisms. At the same time, additional biological processes, like active secretion and phagocytosis, may also be involved [4,10,11,12]. It can be found in many biological fluids, and its levels are altered by many pathological and physiological conditions, such as physical activity or sepsis [4]. Particularly, cancer-bearing patients have more plasma cfDNA in the bloodstream than healthy individuals. Subtracting the amount of cfDNA originated in healthy cells and in cells from the tumour microenvironment under hypoxia, only a small percentage of cfDNA originates from tumour cells and is considered circulating tumour DNA (ctDNA), varying between 0.01 and 93% of all plasma cfDNA [4,10,13].

The quantity of plasma ctDNA differs according to the type and location of the tumour, the stage of the disease, and the treatment received by the patient [10,14]. Despite evidence of a lower ctDNA shed from peritoneal metastasis of gastrointestinal cancers, when compared with visceral metastasis [15], ctDNA can be easily obtained in samples of ovarian cancer, particularly in those with advanced-stage disease [11,14,16,17]. Also, genetic alterations may influence ctDNA release. In *TP53* and *KRAS*-mutated lung cancer, higher ctDNA rates were detected, possibly due to higher tumour aggressiveness or cellular turnover [11,18]. *TP53* mutations are the defining feature of the most common type of ovarian cancer—HGSOC—and they may help explain, together with the frequently advanced stage of the disease, the literature reports of high ctDNA detection in HGSOC compared with other histological subtypes of OC, ranging from 75 to 100% of all cases [4,14].

cfDNA has a variable half-life depending on the amount shed and clearance capability of metabolizing organs, like the liver and spleen, and plasma circulating enzymes, ranging from 16 min to 2.5 h [10,11,12]. Whilst this characteristic challenges cfDNA sample collection and analysis, it allows for real-time assessment of tumour genetic characteristics. Also, in cancer patients, impaired cfDNA clearance results in higher ctDNA concentrations in the bloodstream [10].

cfDNA includes coding and non-coding nuclear and mitochondrial DNA (mtDNA) and generally ranges from 40 to 200 bp in size [10,11]. These shorter fragments originate through caspase-dependent cleavage during apoptosis, with a peak at 160 bp corresponding to nucleosomes [10,11]. Shorter fragments (<100 bp) seem to be enriched in ctDNA, carrying tumour-driven genetic alterations. Conversely, longer cfDNA fragments (>200 bpm) suggest more DNA integrity and are more frequent in cancer patients, possibly because cancer cells die more regularly from necrosis and autophagy than healthy tissues, especially in advanced tumours [11]. Enrichment by size, but also according to nucleosome positioning and the cleavage site nucleotide motifs of cfDNA, may therefore augment ctDNA study outputs [11,19,20].

Theoretically, ctDNA represents a picture of the genetic patrimony of the tumour, corresponding to all metastatic niches, that can be studied and compared longitudinally through time [11,13]. Due to its multiple origins, simple cfDNA quantification is non-informative, and identification and quantification of ctDNA require a qualitative analysis, which can be informative in itself [1,10]. Tracked qualitative changes can be genomic alterations, which include insertions/deletions (indels), single nucleotide variants (SNVs), gene fusions, and copy number alterations (CNAs), but also preferred DNA end motifs, chromosomal instability, DNA methylation profiles, and nucleosome footprints [10,11]. Quantitative and digital polymerase chain reaction (PCR) and targeted and non-targeted next-generation sequencing (NGS) are common approaches for ctDNA quantification [10,13]. While PCR focuses on a few known genomic markers, non-targeted NGS allows for the identification of multiple known or new alterations, allowing for a more comprehensive analysis and improved diagnostic capacity [21,22].

Circulating tumour DNA has been shown to have potential applicability in numerous settings for OC clinical management. While sequencing may be an expensive and unnecessary method for tracking ctDNA quantitative changes during cancer treatment, sequencing-based approaches may uncover potentially new biomarkers with different predictive properties (Figure 1) [8]. In this review, we will explore the most recent data on ctDNA as a biomarker of OC. With NGS emerging as a standard procedure for treatment selection for many solid neoplasms, we will focus on the NGS methodologies used for ctDNA analysis, their evolution, challenges, and potential clinical applicability in liquid biopsies of ovarian cancer.

## 2. Choosing NGS for ctDNA Detection and Quantification

NGS of cfDNA has two main goals: to detect the presence of ctDNA with the highest sensitivity and specificity possible and to characterise the tumour genome [21]. The most commonly used sequencing approaches to study cfDNA in OC patients are targeted sequencing, whole exome sequencing (WES), and whole genome sequencing (WGS) [4]. Targeted sequencing focuses on specific regions of the genome that can be predefined or personalised according to the results obtained by sequencing the primary tumour or baseline plasma samples, allowing mainly the detection of SNVs and indels [4,12]. In targeted sequencing, enrichment is applied to select regions of interest against the whole genomic background, making it more accurate and reducing costs [23]. Enrichment can be amplicon-based, like tagged-amplicon deep sequencing (TAm-Seq), or hybrid capture-based, using biotinylated oligonucleotide probes that hybridize with the regions of interest [4,21,23]. WGS and WES are more comprehensive techniques, targeting the whole genome or only the protein-coding genes, respectively [21]. They produce more data and allow the study of different genetic alterations, including the detection of CNAs, rearrangements, and chromosomal abnormalities [12].

To detect ctDNA, digital droplet PCR may be used, but it requires a pre-identified genetic target [4,24]. With a higher cost, NGS allows for the analysis of multiple somatic or germline genetic alterations, thus allowing for ctDNA detection based on the identification of multiple targets simultaneously [4,22]. A commercially available platform can be used, or it can be created based on a comprehensive genome study (WES or WGS) of the solid tumour (tumour-informed), which allows for a more selective strategy of ctDNA detection through a single sequencing process. This is of particular importance when the quantity of cfDNA is a limiting factor [4,21].

Cancer Personalised Profiling by Deep Sequencing (CAPP-Seq) is a less expensive and highly sensitive method first implemented for non-small cell lung cancer. It uses WES data from 407 tumours to build a targeted sequencing platform to cover the most frequently altered 521 exons and 13 introns [25]. Pereira et al. (2015) used a similar approach with digital PCR (dPCR) in OC. Combining somatic NGS for identifying genetic alterations in 22 ovarian carcinomas (21 of serous histology), ctDNA quantification by NGS-guided dPCR yielded a high diagnostic sensitivity (99%) and specificity (81%) for OC, accompanied by a ctDNA detection rate superior to 93% [4,26]. dPCR is as sensitive and more cost-effective than molecular barcoded NGS in the detection of *TP53* mutations in longitudinal cfDNA if a tumour-informed approach is used [24].

Considering the use of NGS, the limit for ctDNA detection varies between 0.03% and 1% and depends on numerous factors: the quantity of ctDNA, the number of genetic alterations tracked by the test, the coverage of each test for a specific locus, and the technology used [4,21]. The percentage of mutant variant reads, considering the total number of reads in a sample, designated by variant allele frequencies (VAFs), is influenced by the amount of ctDNA available, which varies according to tumour burden and stage [10,21]. For this reason, in order to be generally applicable, detecting even low VAFs (<1%) is a requirement of NGS-based platforms [21].

Increasing the number of targeted regions, or conducting WES or WGS (i.e., increasing the breadth of the platform), increases the probability of encountering a mutated fragment of tumoral origin among all cfDNA fragments in an obtained plasma sample [21]. Coverage, or depth, refers to the number of reads that align with or “cover” a specific nucleotide sequence. Therefore, higher depth also increases the confidence in detecting a particular alteration in a specific locus and improves the detection of rare variants caused by tumour heterogeneity (low cellular representation of a specific clone) or low ctDNA levels [21].

In all stages of cancer, including the early stages, when the number of ctDNA fragments in the plasma is lower, targeted sequencing ctDNA detection is limited to a tumour fraction (TF) of 10^−3^, with TF being the percentage of cfDNA that is ctDNA, according to the NGS output [27]. These results were obtained in 200 patient cfDNA samples, 42 of which had ovarian cancer, using a targeted error correction sequencing (TEC-Seq) methodology [27,28]. Despite covering only 55 genes, the amplicon-based TEC-Seq method uses a predefined small number of barcodes (molecular identifiers) to identify each molecule of ctDNA and redundant sequencing to reduce sequencing errors, allowing for the highest sensitivity (75–100%) and specificity (>80%) in OC diagnosis reported for a targeted NGS platform [4,10,28,29]. These unique molecular identifiers (UMI), or barcodes, allow for, after amplification, grouping and comparison of all reads of amplicons with origin in the same ctDNA molecule and, therefore, the exclusion of different reads identified as errors [30].

UMI-tagged NGS is particularly helpful in samples with low TF, yielding a more sensitive detection of genetic alterations, like *TP53* mutations in HGSOC [30,31]. Likewise, increasing the depth of such platforms can yield a detection limit of up to 10^−5^ TF, especially important in evaluating residual disease (RD) or early-stage cancers [21,26,27]. In this study by Asaf Zviran et al. (2020), WGS of the solid tumour (genome-wide mutational integration) identified thousands of somatic alterations, allowing for a higher number of informative genetic markers to be present in the targeted-NGS platform used for ctDNA detection [27]. While comprehensive gene approaches, like WES, may be applicable to studying ctDNA directly, it requires lots of resources, making personalised and tumour-informed targeted NGS techniques, like CAPP-seq, a more cost-effective methodology for detecting and qualitatively accessing ctDNA, especially in sequential longitudinal analyses [21,26].

Together with the evolution of sequencing techniques, new bioinformatic tools based on artificial intelligence (AI) may help to overcome the main issues of ctDNA analysis. Several tools have been created to allow variant calling at low TF, such as DeepVariant, Clairvoyante, and MRD-EDGE [32,33,34]. The performance of the tools varies depending on fragment length and the type of variant to be analysed, but they seem to outperform the conventional platforms [35]. AI-based variant calling helps to lower the costs and complexity of NGS protocols and yields robust results even with low sample quantities, which is of particular interest during neoadjuvant treatment and in detecting minimal residual disease. However, the clinical applicability of these tools needs to be determined within clinical trials [32].

## 3. NGS-Based Prognostication and Prediction of Treatment Response

There are many studies evaluating the prognostic impact of ctDNA alterations in OC; however, many of them used platforms with limited coverage. Makoto Nakabayashi et al. (2018) used a commercial non-invasive prenatal testing low-coverage WGS to study the ctDNA of 36 OC patients. They found that the presence of copy number alterations (CNAs) in cfDNA was related to worse overall survival (OS) and progression-free survival (PFS). With low coverage, CNAs were only identified in 6 out of 36 patients and only in 1 out of 21 patients with early-stage OC. Also, the origin of CNAs was not confirmed by the sequencing of the primary tumour [36]. In six patients with HGSOC, the same methodology also showed CNAs in specific genes of patients resistant to neoadjuvant chemotherapy (NACT), which significantly impacted PFS [37].

Selecting therapies and accessing treatment responses based on ctDNA changes remains a challenge without known clinical significance, particularly in cancers without dominant oncogenic mutations like OC. Analysing with a 500-gene NGS platform 78 longitudinal ctDNA samples from 12 patients with HGSOC, collected at random time points, with a high concordance with tumour samples (79%), Jaana Oikkonen et al. (2019) reported that patients with a good response to chemotherapy present higher proportions of mutations with decreasing VAF compared with poor-responding patients. Also, they were able to detect potentially actionable mutations in 58% of the included patients [38]. A CAPP-seq platform also showed an increasing number of variants in NACT-resistant patients when compared with the sensitive ones [39,40]. Therefore, it is possible to monitor patients during NACT using ctDNA, and tumour mutational burden (TMB) may be informative of patients’ chemosensitivity [40].

While ctDNA may help to define outcomes in the neoadjuvant setting, it may also directly inform the choice of treatment in advanced non-operable OC patients. The first part of the TARGET study showed that, considering a VAF threshold of 2.5%, actionable mutations were found in the ctDNA of 41 out of 100 patients with advanced cancers [41]. OC was underrepresented, and only the second part of this study, which is ongoing, will focus on the effect of matching therapies with cfDNA findings [41,42]. However, the selection of therapy matched by cfDNA alterations appears to be an independent factor in better outcomes in gynaecological cancers [43]. In a subset analysis of the PERMED-01 study with plasma samples from 24 heavily pretreated OC patients, NGS of ctDNA resulted in the identification of potentially actionable mutations in 42% of patients, of which 3 received targeted therapy, with a median PFS of 4.83 months (0.66–25.26) [44,45].

Mechanisms for PARP inhibitor (PARPi) resistance are under study; however, *BRCA1* and *BRCA2* secondary mutations resulting in re-expression of functional or partially functional BRCA1 and BRCA2 proteins are a well-known resistance mechanism to PARPi and platinum-based chemotherapy that can be studied using cfDNA [46,47,48,49]. Targeted sequencing assays have shown that detection of BRCA secondary mutations in cfDNA is more common in platinum-resistant or refractory HGSOC, compared with those with platinum-sensitive disease, and predicts a worse outcome and decreased PFS in patients receiving PARPi treatment, namely rucaparib [46]. However, multiple and nonexclusive mechanisms and genetic alternations may be responsible for PARPi resistance, despite their unknown clinical relevance. Other mechanisms than BRCA secondary mutations for homologous recombination repair (HRR) restoration, replication fork stabilization, and survival signal upregulation are also potentially accountable. Those mechanisms were found to be present post-PARPi treatment in the ctDNA of 89.7% of OC patients when compared with pre-treatment samples. Also, patients with HRR restoration performed worse in terms of PFS following subsequent chemotherapy treatments, as well as those harbouring two or more of these resistance mechanisms at once [47]. Secondary mutations in HRR genes like RAD51C and RAD51D post-PARPi treatment are further examples of resistance mechanisms that may also lead to the restoration of HHR [50].

Despite the potential interest of qualitative assessment of ctDNA to evaluate prognosis during systemic treatment, many studies showed that ctDNA identification and quantification with longitudinal collections can help predict disease recurrence, outperforming the currently used CA-125 [6,8,51,52]. Additionally, quantitative data may also be helpful to evaluate residual disease and prognosis in many tumour types [21]. Figure 2 illustrates the potential clinical value of ctDNA quantification in different OC clinical scenarios.

Angel Chao et al. (2022) found that detection of ctDNA after primary cytoreduction or interval debulking surgery in OC predicts a worse outcome regarding PFS and OS, while ctDNA levels at diagnosis had no prognostic significance [53]. With a 59-gene panel, Zhu et al. (2023) reached a similar conclusion: post-surgical detection of ctDNA could be predictive of a worse prognosis [54]. Despite the vast majority of authors finding a relationship between ctDNA presence and worse outcomes, the results are not consistent between studies. In a slightly different study of two cohorts of 25 patients enrolled after cytoreductive surgery with or without subsequent adjuvant chemotherapy (ACT), ctDNA detection post-ACT is a significant predictor of worst recurrence-free survival (RFS), while ctDNA levels after surgery are not [52]. Nonetheless, a rise in ctDNA levels during the first cycle of chemotherapy may correlate with greater cancer cell destruction and chemosensitivity, therefore being a potential marker of a good prognosis, even in recurrent OC [55]. Different sequencing protocols and heterogeneous characteristics of the included patients can explain the incompatible results between studies.

Regarding pre-operative ctDNA, conflicting results were also found by Lara Paracchini et al. (2021), with baseline high TF predicting worse PFS at a cut-off point of 6.5%, but no validation set was applied to confirm these findings [51]. Also, June Y. Hou et al. (2022) showed that patients who progressed and died in his cohort had higher baseline ctDNA [52]. Including all subtypes of epithelial OC and using a targeted NGS platform, despite being tumour-informed, these observations from June Y. Hou et al. (2022) may result from differences in the assay sensitivity for different types of tumours. However, these findings agree with other non-NGS-based evidence, showing a positive and independent correlation between pretreatment ctDNA and OS or PFS [56].

In more advanced stages of OC, namely in platinum-resistant patients receiving bevacizumab treatment, ctDNA levels appear to be independent predictors of OS and PFS [9,57]. Published results from plasma WES and low coverage WGS in OC of patients with a median of three previous lines of systemic treatment, a majority of which were platinum-resistant, showed that TMB and genome altered fraction (i.e., sum of copy number alteration divided by the total number of studied regions after removing sex chromosomes) are also inversely correlated with PFS [45]. These results show that by applying NGS technologies, particularly with high breadth, it is possible to obtain other study variables, like the above-mentioned TMB, that may yield important prognostic information [21]. Additionally, they show the utility of ctDNA quantification applied to different points of OC evolution, including heavily treated and platinum-resistant patients, a setting with more limited data [9,45].

With longitudinal cfDNA collection, the ctDNA genomic profile could be informative for the selection of the next line of treatment. Also, with routine collections, molecular recurrence can be detected and used to anticipate clinical and radiological recurrence by months [26,51,52,58,59]. However, the low sample size and unstandardized methods of collection in existing studies impact the clinical applicability of the results [51,52,58].

## 4. The Challenge of Tumour Clonality

Theoretically, when accessing the tumour genome, particularly in relapsed or pretreated disease, ctDNA analysis should help overcome the issues of clonality within tumours, spatial heterogeneity between different tumour locations, and temporal evolution of tumour genetic background according to treatment-induced selective pressure [58,60]. Many studies have focused on studying ctDNA concordance with the tissue sample. Most were performed in HGSOC using PCR analyses of the *TP53* mutation, since more than 90% present this genetic alteration [4]. This technique yields high detection rates of mutant ctDNA, with high sensitivity (>75%) and specificity (>80%) when compared with the tumour genomic information [4,14].

Overall, Ana Barbosa et al. (2021) showed that at least one somatic variant presented in the tumour could be found in the ctDNA of 35.6% of patients, and the concordance increased to 69.6% for patients not previously treated and 83.3% for those presenting stage IV disease [61]. Using NGS to compare cfDNA and multiple and simultaneously retrieved HGSOC biopsy samples, ctDNA presents more SNVs with a low concordance (~2%), while exhibiting higher concordance in the number of CNAs, which was mostly above 50% for each patient [58]. These findings may suggest that ctDNA represents a better snapshot of the tumour’s molecular characteristics and clonality than the biopsy and the imbalances explained by low VAF variants, which may be undetected, particularly when cfDNA TF is low [58,61].

In OC and endometrial cancers, *PTEN*, *KRAS*, *BRCA2*, *BRCA1*, *GNAS*, *PIK3CA*, *ERBB2*, *ARID1A*, and *TP53* had the highest and most significant concordance rates in a series [62]. Overall concordance rates in gynaecological malignancies range from 75.6% to 88.5% for *KRAS*, *TP53*, and *PIK3CA* alterations. However, these results seem not to be influenced by the location of the biopsy (primary tumour or secondary location) or the length of time between the biopsy and cfDNA collection [43]. A large multi-tumour cohort showed that higher agreement between ctDNA and matched tumour genomes was obtained for SNVs and CNAs of truncal gene alterations, while low concordance was often found for genes involved in treatment resistance mechanisms. One can hypothesise that these disparities may reflect the clonality within the tumour, which is mostly due to acquired alterations that cause treatment resistance rather than alteration of the main oncogenic drivers [62]. In a reverse methodology, Ana Barbosa et al. (2021) studied secondary tumour location in search of variables detected in cfDNA only. In one out of five patients, they could find the same variant in cfDNA and an ovarian mass contralateral to the primary tumour. Because both ovarian masses shared 2 out of 3 rare variants identified in ctDNA, the authors hypothesise that, despite being physically distant and genetically different, the two ovarian masses have a clonal origin [61].

Treatment resistance emerges through clonal selection but may also be caused by polyclonal mechanisms induced by targeted treatments and synthetic lethality-based therapies. In 21% of *BRCA1/2* germline mutated patients, the cfDNA of platinum and/or PARPi-treated OC showed the presence of polyclonal putative *BRCA1* or *BRCA2* secondary mutations at low VAF, which increased after PARPi treatment and resistance. Moreover, 4 out of 6 patients with reverse mutations presented with more than one concurrent *BRCA1* or *BRCA2* variant, particularly in *BRCA2* [48]. Ensuring the importance of cfDNA in studying resistance mechanisms, clonality will probably not explain the phenomenon on its own, as treatment resistance is likely a multifactorial event [48].

## 5. Methylation Patterns in ctDNA

Methylation is a relatively stable modification of DNA that occurs in cytosine–phosphate–guanine (CpG)-rich regions (CpG islands) and alters the expression of certain genes. In cancer, changes in the expression of tumour suppressor genes by methylation of their promotor are an early tissue- and cancer-specific event [63]. In OC, several genes are commonly silenced through methylation, like *ARH1*, *BRCA1*, *CDKN2A*, *MLH1*, and *RASSF1* [17,64]. Methylation patterns in OC have also been unveiled using ctDNA, mostly through PCR-based studies like methylation-specific PCR (MSP), real-time methylation-specific PCR (RT-MSP), multiplex nested methylated specific PCR (MN-PCR), or other methodologies such as microarray-mediated methylation assay (M3-assay) [4]. Hypermethylation of tumour suppressor genes in ctDNA, like *RASSF1a* and *BRCA1*, has been explored as a non-invasive biomarker in OC due to its stability across different stages and grades, with potential applicability in diagnosis [64,65,66,67].

The use of sequencing technologies to study methylation in OC has been growing, as has its application to cfDNA. Simone Karlsson Terp et al. (2023) reported in a systematic review, including 29 original studies of OC early detection in cfDNA, that a targeted bisulfite NGS, applied to 39 OC patients and covering 1,116,720 CpGs, performed the best to discriminate OC patients from healthy controls, with a sensitivity of 100% and a specificity of 99.3% in a validation cohort of 12 patients [63,68]. Martin Widschwendter et al. (2017) had inferior results with the same technology but using a 3-gene methylation-serum-marker panel (*COL23A1*, *C2CD4D*, and *WNT6*), with a sensitivity of 41.4% and a specificity of 90.7% in a validation cohort of 41 patients, stage I to IV [7]. These inferior results may be explained by the restricted panel used, but also because the patients had different histologic types of ovarian carcinoma, whereas controls were composed of other cancers, benign conditions, and non-epithelial and borderline OC [7,63].

Nevertheless, panels, instead of methylation biomarkers for single genes, seem to perform better for OC early diagnosis, hence the interest in NGS technologies in this setting [63]. With a different sequencing approach, pyrosequencing, Dana Dvorská et al. (2019) analysed the methylation profiles of *RASSF1*, *PTEN*, *CDH1*, and *PAX1*. Their results showed increased methylation of *PAX1* in OC compared with controls and a high concordance between tumour and ctDNA samples. The methylation profile of these four genes, related to cell proliferation and differentiation, combined with patients’ age in a predictive model allowed the differentiation between controls and OC in plasma samples with 91% sensitivity and 56% specificity [69].

Thus, evidence has shown the prognostication capability of OCs methylation profile [69,70,71]. Leilei Liang et al. (2022) built a prognostic and detection model based on methylation sequencing of HGSOC ctDNA and benign ovarian diseases [72]. The diagnostic model included 18 genes and distinguished OC from benign ovarian conditions and healthy controls with 94.7% sensitivity and 88.7% specificity, with efficiency increasing with the tumour stage and outperforming CA-125. The prognostic model included 15 methylation markers and divided the patients into high- and low-risk groups. High-risk patients had shorter PFS independently of the tumour stage and were more platinum-resistant. It also had better performance in prognostication than CA-125. Using the Cancer Genome Atlas Program cohort, the authors found a higher homologous recombination deficiency (HRD) score in low-risk patients, while the HRR pathway genes had a balanced frequency of mutations between high and low-risk, suggesting that HRD variance may not be dependent on genomic information from HRR genes but instead on epigenetic modulation. Therefore, low-risk patients may also have a decreased capability of immune evasion and higher immunogenicity, which may contribute to better outcomes [72].

However, methylation influences treatment outcomes, and ctDNA methylation patterns may be used not only for early diagnosis but also to track treatment response and disease recurrence. Collected paired samples of 25 patients undergoing NACT chemotherapy and 23 with HGSOC showed that the levels of the three methylation markers dropped with exposure to platinum-based chemotherapy and performed better than the CA-125 cut-off of 35 IU/mL to accurately identify responders (in 78% of cases) and non-responders (in 86% of cases) [7]. Similarly, Tushar Tomar et al. (2017) used genome-wide DNA methylation-enriched sequencing (MethylCap-seq) to study extreme and non-responders in HGSOC. They found four epigenetic signatures related to treatment response: *FZD10* and *MKX* hypermethylation in extreme responders, and *FAM83A* and *MYO18B* hypermethylation in poor responders. Also, in an external validation cohort and another in silico, the authors demonstrated that both *FZD10* and *MKX* hypermethylation lead to better PFS and OS. *FZD10* expression, a WNT pathway receptor, is also correlated with survival outcomes, and its downregulation increases cisplatin sensitivity, as the authors demonstrated in functional studies using SKOV3 and OVCAR3 cell lines [73]. Despite the crucial role of cell lines in identifying potentially relevant methylation profiles and their molecular impact on tumour development and treatment response, results in these models may not translate to the solid tumour or the clinical phenotype [74,75,76].

Methylation has a tight relationship with cancer treatments, changing with exposure to different drugs and altering the pattern of cancer cell responses to systemic therapies [7,71,77]. In OC, epigenetic regulation of *BRCA1* may influence PARPi and platinum sensitivity, and methylation of *BRCA1* appears mutually exclusive with mutations of the same gene [78,79]. Methylation of the *BRCA1* promoter is an exclusive event in cancer cells, which makes its detection in ctDNA highly sensitive. Maha Elazezy et al. (2021) showed that, at some point in HGSOC evolution, 60% of patients present this epigenetic alteration, although 24% of them lost hypermethylation during treatment. These findings may refer to an active mechanism of resistance or to clonal methylation patterns in HGSOC [79].

While *BRCA1* methylation is one of the main causes of *BRCA1* dysfunction in HGSOC non-BRCA-mutated patients, the impact on survival outcomes of this epigenetic modulation is yet to be fully understood; however, *BRCA1* methylation was shown to be predictive of response to an iPARP named rucaparib [78,80,81]. *HOXA9* methylation is also detectable in ctDNA and can occur during PARPi treatment, correlating to worse OS and PFS [67,82]. Other gene methylation levels can indicate ovarian cancer sensitivity to treatment; however, most data arise from cell line studies [78]. While *BRCA1* methylation alone seems an insufficient biomarker to explain the complex treatment resistance in OC, more comprehensive methylation studies, namely NGS-based, may help explain treatment response variability that occurs independently of genomic alterations [79].

## 6. ctDNA in Non-High-Grade Epithelial Ovarian Cancers and Rare Subtypes

OC is a heterogenous disease with multiple histologic subtypes. Epithelial tumours comprise 90% of all cases, and the most frequent subtype is HGSOC [13,14]. Most OC ctDNA studies have been performed in HGSOC because it has an ideal biomarker comparator, CA-125, which is elevated in more than 90% of the cases. However, there is evidence of a lower concordance between the mutations identified through targeted sequencing in the tumour and ctDNA for other epithelial OC (31.3%) when compared with HGSOC (92.3%). A commercially available targeted NGS panel covering 50 genes was used and showed a higher percentage of detectable mutations in HGSOC at diagnosis compared with other histological subtypes. These data were obtained from a small cohort with 13 HGSOC and 16 OC of other epithelial origin, mainly clear cell carcinomas (CCC) [53].

Leilei Liang et al. (2022) developed a methylation sequencing-based diagnostic model for OC that yields good discriminatory capability from non-cancer patients even at earlier stages of HGSOC. While the model maintained its specificity and sensitivity for other histological types of OC, in exploratory analyses of eight borderline epithelial OCs (4 mucinous, 3 serous, 1 endometrioid, and 1 mixed), the sensitivity dropped to 12.5% [72].

## 7. Conclusions

Allowing for the analysis of tumour genomes transversally to all tumour niches and longitudinally through time, as well as correlating with tumour burden and stage, ctDNA analysis based on NGS technologies emerges as a future clinically relevant biomarker in OC, particularly in HGSOC [1,4]. Despite the higher cost and resource consumption, NGS has the advantage of analysing multiple and different alterations simultaneously in different parts of the genome [24,25]. Also, it allows for ctDNA quantification and the assessment of multiple other variables, like VAF and TMB, that can be used to correlate with clinical information and predict patients’ outcomes [45]. Therefore, mounting evidence suggests that ctDNA may exceed the diagnostic and predictive capabilities of CA-125, the current gold standard biomarker in OC [1,8,51,52].

Advanced stages of OC lack therapeutic options, and cfDNA sequencing may help identify and stratify potential treatment targets for those patients [45,57]. Ideally, longitudinal genetic characterization, but also follow-up of TF, are upcoming tools to select the best treatment option, which patients to treat, or even to personalize treatment duration, reducing associated toxicities and comorbidities [83]. cfDNA seems to allow for treatment follow-up in targeted therapies and chemotherapy, both in the early and later stages of the disease [37,38,45,57].

Identifying low VAFs and rare CNAs in ctDNA and understanding their significance remains the main issue of NGS-based studies, mainly in patients exhibiting a low TF [21,27]. To increase the potential clinical applicability of liquid biopsy, OC protocols need to be optimised to identify ctDNA more efficiently and to be capable of calling for rare alterations that may be informative regarding tumour polyclonality and adaptative mechanisms [48,58,61]. AI-based tools are emerging as valuable assets to overcome these barriers [32,35].

Beyond genomic analyses, multiple target methylation studies have unveiled the polyclonal and multi-mechanistic nature of acquired treatment resistance in OC [48]. Therefore, longitudinal NGS-based cfDNA methylation analysis may be a future independent tool to diagnose OC, track treatment response, and facilitate early prediction of treatment resistance. Also, while ctDNA cumulates evidence of utility in HGSOC, its applicability to other epithelial non-HGSOC and rare OC remains unknown and seems to be heterogeneous between subtypes [53,72].

Data regarding OC liquid biopsy derives mainly from retrospective cohorts with non-standardized liquid biopsy collection time points or small and exploratory prospective samples. In larger studies, OC remains underrepresented, especially rarer subtypes. Given the costs of NGS, ctDNA collections should be included in large prospective cohorts, preferably in the setting of clinical trials.

## Figures and Tables

**Figure 1 biology-13-00088-f001:**
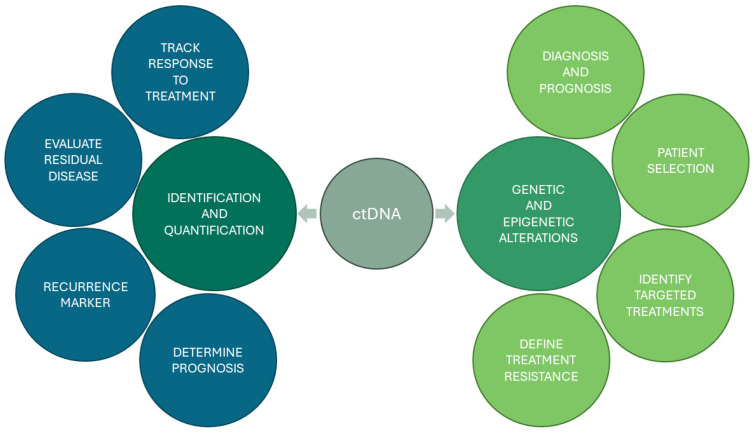
Potential clinical applications of circulating tumour DNA (ctDNA) as a biomarker in ovarian cancer vary according to the type of methodology (quantitative or qualitative).

**Figure 2 biology-13-00088-f002:**
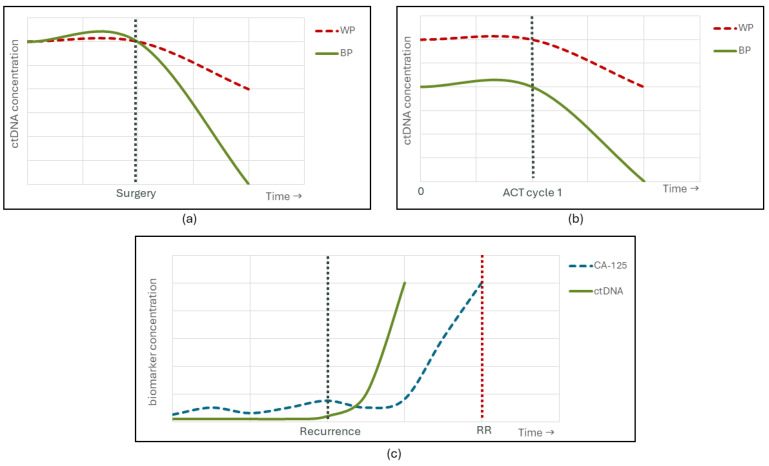
Graphical representations of results of some included studies. (**a**) A patient with detectable ctDNA after primary cytoreduction or interval debulking surgery (red) has worse overall and progression-free survival. (**b**) A patient with higher levels of ctDNA at baseline has a worse prognosis (red). Independently of baseline levels of ctDNA, a patient with detectable ctDNA (red) after the first cycle of adjuvant chemotherapy (ACT) has a worse recurrence-free survival. (**c**) In a recurrence, ctDNA levels (green) are elevated before Ca-125 levels (blue), and both before radiological detection of recurrence (RR). WP—worse prognosis; BP—better prognosis.

**Table 1 biology-13-00088-t001:** Comparison between Ca-125 and ctDNA as biomarkers of OC.

CA-125	ctDNA
Non-invasive	
Can be altered by other coexisting physiological and pathological conditions	
Inexpensive and highly available	Expensive and restricted to specialist centres
Simple methodology	Complex methodology
Results in minutes-hours	Results in days and weeks
Quantitative marker	Quantitative and qualitative markers
One continuous variable	Can generate multiple continuous and discrete variables
Only informative regarding the presence/absence of treatment response and recurrence	Yields more information regarding treatment response and tumour recurrence, like resistance mechanisms and targetable genetic alterations
Directly interpreted by the clinician	Requires specialised interpretation
Easily detected in blood and urine	Low concentrations in biological fluids
The utility is limited to producing tumours (mainly restricted to HGSOC)	Theoretically applicable to all histological subtypes
Established and recognised clinical utility in trials	Clinical utility is debatable and requires confirmation in prospective trials

HGSOC—High grade serous ovarian cancer.

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
