# Peer review of "Current Applications and Challenges of Next-Generation Sequencing in Plasma Circulating Tumour DNA of Ovarian Cancer"

_biology, 2024, doi:10.3390/biology13020088_

Round 1
Reviewer 1 Report
Comments and Suggestions for Authors
The manuscript needs following improvements before publication,
@ Provide more context for why the study is significant. Why is understanding ctDNA in ovarian cancer important, and how does it contribute to the existing body of knowledge in the field?
@ The manuscript have no figures, while for visual illustration its really improtant. Following are suggested diagrams that can be part of the manuscript,
- Schematic representations of ctDNA evolution during treatment - Comparative figures highlighting the concordance between ctDNA and tissue tumor biopsy, particularly in advanced ovarian cancer - Flowcharts illustrating the potential applications of ctDNA in prognostic prediction and treatment selection - Graphs presenting quantitative data on copy number alterations and gene variants during treatment
@ As the manuscript is more near to Review article therefore, I feel like the manuscript should include a section about in silico techniques used in next-generation sequencing. I suggest following literature papers to be included,
- ORI-Explorer: a unified cell-specific tool for origin of replication sites prediction by feature fusion - Machine learning guided signal enrichment for ultrasensitive plasma tumor burden monitoring - DL-m6A: Identification of N6-methyladenosine Sites in Mammals using deep learning based on different encoding schemes - GR-m6A: Prediction of N6-methyladenosine sites in mammals with molecular graph and residual network
Please carry out your own search and include more such latest manuscripts that are used for next-generation sequence analysis.
Comments on the Quality of English LanguageThats fine
Reviewer 2 Report
Comments and Suggestions for Authors
In the manuscript, the authors report that they reviewed the current applications and challenges of NGS in plasma circulating tumour DNA of ovarian cancer. The authors explored the existing data on ctDNA as a biomarker of OC. The review provides reference for the liquid biopsies of ovarian cancer. But they still need to be improved as mentioned below.
1. Please add relevant charts to assist reading.
2. What advantages can NGS technology bring to the application of ctDNA?
3. What is the reason for the inconsistency between the results of ctDNA and related studies?
4. Please discuss the advantages and disadvantages of ctDNA as compared to the ideal biomarker comparator CA-125 in HGSOC.
5. The authors can explore the current difficulties and problems in applying ctDNA to other epithelial non-HGSOC and rare OC.
6. Whether cell lines studies have implications for the study of ctDNA methylation patterns?
Round 2
Reviewer 1 Report
Comments and Suggestions for Authors
Thanks for addressing the concerns.
Comments on the Quality of English LanguageThats fine